# The Emerging Role of Histone Deacetylase Inhibitors in Cervical Cancer Therapy

**DOI:** 10.3390/cancers15082222

**Published:** 2023-04-10

**Authors:** Iason Psilopatis, Nikolaos Garmpis, Anna Garmpi, Kleio Vrettou, Panagiotis Sarantis, Evangelos Koustas, Efstathios A. Antoniou, Dimitrios Dimitroulis, Gregory Kouraklis, Michail V. Karamouzis, Georgios Marinos, Konstantinos Kontzoglou, Afroditi Nonni, Konstantinos Nikolettos, Florian N. Fleckenstein, Christina Zoumpouli, Christos Damaskos

**Affiliations:** 1Department of Gynecology, Charité—Universitätsmedizin Berlin, Augustenburger Platz 1, 13353 Berlin, Germany; 2Second Department of Propedeutic Surgery, Laiko General Hospital, Medical School, National and Kapodistrian University of Athens, 11527 Athens, Greece; 3Nikolaos Christeas Laboratory of Experimental Surgery and Surgical Research, Medical School, National and Kapodistrian University of Athens, 11527 Athens, Greece; 4First Department of Propedeutic Internal Medicine, Laiko General Hospital, Medical School, National and Kapodistrian University of Athens, 11527 Athens, Greece; 5Department of Cytopathology, Sismanogleio General Hospital, 15126 Athens, Greece; 6Molecular Oncology Unit, Department of Biological Chemistry, Medical School, National and Kapodistrian University of Athens, 11527 Athens, Greece; 7Department of Surgery, Evgenideio Hospital, Medical School, National and Kapodistrian University of Athens, 11527 Athens, Greece; 8Department of Hygiene, Epidemiology and Medical Statistics, Medical School, National and Kapodistrian University of Athens, 11527 Athens, Greece; 9First Department of Pathology, Medical School, National and Kapodistrian University of Athens, 11527 Athens, Greece; 10Obstetric and Gynecologic Clinic, Medical School, Democritus University of Thrace, 68110 Alexandroupolis, Greece; 11Department of Diagnostic and Interventional Radiology, Charité—Universitätsmedizin Berlin, Augustenburger Platz 1, 13353 Berlin, Germany; 12Berlin Institute of Health, Charité—Universitätsmedizin Berlin, BIH Biomedical Innovation Academy, BIH Charité Clinician Scientist Program, 13353 Berlin, Germany; 13Department of Pathology, Sismanogleio General Hospital, 15126 Athens, Greece; 14Renal Transplantation Unit, Laiko General Hospital, 11527 Athens, Greece

**Keywords:** histone, deacetylase, inhibitor, cervical, cancer, therapy

## Abstract

**Simple Summary:**

Histone deacetylase inhibitors (HDACIs) are a relatively new drug class with important effects on the epigenetic regulation in cancer, inducing cancer cell death, apoptosis induction, and cell cycle arrest. Even though HDACIs have, to date, received approval for mainly hematologic malignancies, there are plentiful studies in cervical cancer setting with encouraging results. The present review summarizes all studies with HDACIs in cervical cancer from the emerging data in labor research to the possible applicability in clinical practice.

**Abstract:**

Cervical carcinoma is one of the most common cancers among women globally. Histone deacetylase inhibitors (HDACIs) constitute anticancer drugs that, by increasing the histone acetylation level in various cell types, induce differentiation, cell cycle arrest, and apoptosis. The aim of the current review is to study the role of HDACIs in the treatment of cervical cancer. A literature review was conducted using the MEDLINE and LIVIVO databases with a view to identifying relevant studies. By employing the search terms “histone deacetylase” and “cervical cancer”, we managed to identify 95 studies published between 2001 and 2023. The present work embodies the most up-to-date, comprehensive review of the literature centering on the particular role of HDACIs as treatment agents for cervical cancer. Both well-established and novel HDACIs seem to represent modern, efficacious anticancer drugs, which, alone or in combination with other treatments, may successfully inhibit cervical cancer cell growth, induce cell cycle arrest, and provoke apoptosis. In summary, histone deacetylases seem to represent promising future treatment targets in cervical cancer.

## 1. Introduction

The cells lining the uterine cervix represent the cells of origin for cervical cancer, with cells in the exocervix giving rise to squamous cell carcinomas and adenocarcinomas developing from the mucus-producing endocervical gland cells [1]. Most cervical cancers are squamous cell carcinomas and begin in an area where the exocervix joins the endocervix which is referred to as the “transformation zone” [1]. For 2023, the American Cancer Society estimates the incidence of invasive cervical cancer at about 13,960 new cases and the associated deaths at about 4310 new cases in the United States [2]. Despite once embodying one of the most common causes of cancer death in women, cervical cancer, nowadays, accounts for significantly fewer deaths owing to the vaccination against human papillomavirus (HPV), along with the increased employment of screening technology including the Papanicolaou (PAP) and the HPV tests [2]. Nevertheless, cervical pre-cancers still represent a frequent diagnosis described as adenocarcinoma in situ, dysplasia, squamous intraepithelial lesion (SIL), or cervical intraepithelial neoplasia (CIN) [1]. The key risk factor for cervical (pre-) cancer is infection by the high-risk HPV types, followed by chlamydia contamination, immune deficiency, smoking, chronic application of oral contraceptives, multiple full-term pregnancies (and especially a young age at the first full-term pregnancy), a diet low in fruits and vegetables, socioeconomic status, diethylstilbestrol consumption, or use of an intrauterine device (IUD) [3]. While women with pre-cancers or early cervical cancers usually do not show any symptoms, larger invasive tumor masses may cause abnormal vaginal bleeding, unusual vaginal discharge, dyspareunia, or pain in the pelvic region [4]. Treatment options vary by cervical cancer stage and range from surgical approaches (e.g., radical trachelectomy) to radiation therapies (e.g., external beam radiation therapy) with or without chemotherapy, depending also on the wish of mostly young patients to eventually maintain their fertility. More precisely, surgical treatment reaches from conisation via radical trachelectomy/hysterectomy up to exenteration. Standard treatment for locally advanced cervical cancer is platinum-based chemoradiation and for primary metastatic disease primary chemotherapy (+/− immunotherapy) [5].

Efficient packaging of the genetic material in a denser form is a prerequisite for the DNA to fit within the eukaryotic nucleus. This DNA structural organization is enabled by the nucleosome, a positively charged histone octamer containing two identical copies of each of the four core histone proteins H2A, H2B, H3, and H4 [6,7,8,9]. With the negatively charged DNA strands being wrapped around the histones, levels of acetylation on the lysine residues of their aminoterminal tails remain low and the assembly of the basal transcriptional factors is hindered. Consecutively, the preinitiation complex that would facilitate genetic expression may not form [10,11]. Transcription only takes place after post-translational acetylations of the NH2-terminal tails of histones which alter histone affinity for the negatively charged DNA [12,13,14,15,16]. These post-translational acetylations are regulated by the rival effects of histone acetylases (HATs) and histone deacetylases (HDACs) [17,18,19,20] (Figure 1). Of note, HDACs catalyze the removal of acetyl groups on the NH2-terminal lysine residues, thereby repressing transcription and tumor-suppressor gene activation [21,22,23,24]. As a consequence, the deregulation of histone acetylation potentially justifies the genesis of diverse human cancer entities [25,26,27].

Histone deacetylase inhibitors (HDACIs) increase histone acetylation levels in different cell types by inhibiting their target enzymes HDACs and are, thus, evidently involved in carcinogenesis suppression [28,29,30] (Table 1).

Given their potent anticancer effects, diverse HDACIs are currently under investigation in multiple ongoing clinical trials for the treatment of solid cancer entities. So far, only a few HDACIs have, however, received Food and Drug Administration (FDA) approval for T-cell lymphoma or multiple myeloma [31,32]. The present review extensively analyzes the role of HDACIs as promising therapeutic agents for cervical cancer therapy. The literature review was conducted using the MEDLINE and LIVIVO databases. Solely original research articles and scientific abstracts written in the English language, that explicitly reported on the role of HDACIs in cervical cancer were included in the data analysis. Studies focusing merely on the role of HDACs in cervical cancer or the application of HDACIs in non-cancerous cervical cell lines were excluded. By employing the search terms “histone deacetylase” and “cervical cancer”, we identified a total of 254 (duplicate records removed) articles published between 1999 and 2023. After the abstract review, 124 records were discarded in the initial selection process. The full texts of the remaining 130 publications were assessed and a total of 95 relevant studies meeting the inclusion criteria and published between 2001 and 2023 were selected for the final literature review. Figure 2 schematically depicts the aforementioned selection process.

## 2. Hydroxamic Acids

### 2.1. Trichostatin A

Trichostatin A, an antifungal antibiotic initially isolated from Streptomyces hygroscopicus, is a potent and specific HDACI [33].

A great number of studies focus on the antiproliferative effects of trichostatin A as monotherapy on cervical cancer cell lines. Liu et al. suggested that trichostatin A induces human cervical cancer cell apoptosis by decreasing DNA-methyltransferase 3B [34], while another study group treated the HeLa and CaSki cell lines with different concentrations of HDACI and discovered that trichostatin A significantly diminished cell viability, protein arginine methyltransferase 5 (PRMT5), transient receptor potential cation channel, subfamily V, member 6 (TRPV6), and p62, levels, but augmented the apoptosis rate, the LC3 II/I ratio and the beclin1, stanniocalcin 1 (STC1), and phosphorylated Jun N-terminal kinase (p-JNK), protein levels [35]. Furthermore, Ma et al. identified the putative zinc transporter gene *LIV1* as a key gene in the context of trichostatin A-mediated cervical cancer cell apoptosis induction [36], while Raju et al. demonstrated that trichostatin A robustly induces *Kallikrein-related peptidase 7* (*KLK7*) mRNA expression and increases occupancy of specificity protein 1 (Sp1) at the proximal *KLK7* promoter in HeLa cells [37]. Moreover, Wu et al. suggested that the human telomerase reverse transcriptase (hTERT) might represent a trichostatin A target with potent anti-apoptotic effects through p21*^waf1^*-dependent and p53-independent pathways [38]. Li et al. studied the effect of trichostatin A in HeLa cells and concluded that this HDACI activates p21^WAF1/CIP1^ expression through the downregulation of c-myc and release of the repression of c-myc from the promoter [39]. Additionally, Yadav et al. treated C-33A cells with trichostatin A and reported an increased chemokine receptor 4 (CXCR4) expression, leading to increased cell adhesion by a paracrine source of stromal cell-derived factor (SDF)-1α [40]. Lee et al. proved that trichostatin A may mediate hypoxia-inducible factor (HIF) 1α stabilization and translocation into the nucleus for the activation of vascular endothelial growth factor (VEGF) promoter by acetylation at K674 under normoxia, thereby facilitating HDACI resistance in HeLa cells [41], whereas Yang et al. demonstrated that the dynamic F-actin rearrangement plays an important role in trichostatin A-induced HeLa cell apoptosis involving mitochondrial membrane potential collapse [42]. You et al. stated that trichostatin A induces HeLa cell apoptosis in a B-cell lymphoma 2 (Bcl-2) in an oxidative stress-dependent manner [43]. Importantly, Sharma et al. highlighted that trichostatin A suppresses the phorbol 12-myristate 13-acetate (PMA)-induced *osteopontin* transcription in human cervical cancer cells, thus inhibiting tumor growth in vivo [44].

Numerous studies have been also published on the synergistic effects of trichostatin A in combination with other, standard or alternative, treatment agents. Danam et al. cultured O^6^-methylguanine-DNA methyltransferase (MGMT)-silenced HeLa S3 cervical cancer cells and proved that trichostatin A effectively induced *MGMT* RNA expression only after combination with 5-aza-2′ deoxycytidine (5-Aza-dC) [45]. Similarly, Jung et al. treated diverse cervical cell lines with trichostatin A and 5-Aza-dC and highlighted the synergistic effect of this combinational treatment on the reactivation of the *adenylate cyclase-activating polypeptide 1* (*ADCYAP1*) gene expression [46]. Given that the *Fanconi anemia complementation group F* (*FANCF*) gene is disrupted in most cervical cancers, Narayan et al. proved that trichostatin A may induce its re-expression in cervical cancer cell lines, hence counteracting their chromosomal hypersensitivity phenotypes after exposure to alkylating agents [47]. Huisman et al. targeted the demethylation inducers ten-eleven translocation (TET) or thymine DNA glycosylase (TDG) and achieved *EPB41L3* re-expression only after combination with trichostatin A [48]. Furthermore, Hernández-Juárez et al. investigated the effects of hindering histone deacetylation on the expression of *SLC5A8* and concluded that trichostatin A and pyruvate may successfully reverse the epigenetic silencing of the *SLC5A8* gene in cervical cancer cells [49]. Lin et al. described a synergistic effect of trichostatin A or vorinostat and bortezomib in both inducing caspase-mediated apoptosis in HeLa cells and significantly more effectively retarding tumor growth in vivo [50], while Chakraborty et al. preincubated HeLa cells with trichostatin A prior to scaffold/matrix attachment region-binding protein 1 (SMAR1) upregulation by curcumin and revealed that HDAC1 activity weakened SMAR1-mediated *E6* transcriptional repression [51]. Igaz et al. investigated the combinational effect of trichostatin A and silver nanoparticles in HeLa cells and reported enhanced DNA targeting capacity, apoptosis induction, as well as inhibition of cervical cancer cell growth or migration [52]. Accordingly, Zhang et al. synthesized palladium nanoparticles which, in combination with trichostatin A, had a synergistic effect on cytotoxicity, oxidative stress, mitochondrial membrane potential, caspase-3/9 activity, as well as the expression of pro- and anti-apoptotic genes in cervical cancer cells [53]. Yu et al. investigated the trichostatin A-mediated modulation of the radio-sensitivity of HeLa cells under hypoxic conditions and demonstrated that this HDACI radio-sensitizes Hela cells under hypoxic conditions by diminishing the expression of HIF-1α and VEGF proteins [54].

Altogether, trichostatin A, alone or in combination with other drugs, represents a potent HDACI in terms of cervical carcinoma therapy.

### 2.2. Suberoylanilide Hydroxamic Acid

Suberoylanilide hydroxamic acid, or vorinostat, is an HDACI that reacts with and blocks the catalytic site of HDACs [55,56].

Many researchers have, to date, studied the effect of vorinostat on cervical cancer cells. Hancock et al. developed a cell-based assay and proved that suberoylanilide hydroxamic acid potentiated Notch intracellular domain (NICD) signaling, hence promoting the assembly of the NICD/CBF1/Suppressor of hairless/Lag-1 (CSL)/Mastermind-like (MAML) transcriptional activation complex [57], while He et al. treated HeLa cells with suberoylanilide hydroxamic acid and, by utilizing a proteomic approach to reveal protein expression changes, visualized a total of nine differentially expressed proteins, with phosphoglycerate mutase 1 (PGAM1) showing significant downregulation [58]. Moreover, Pan et al. described that vorinostat reverses epithelial-mesenchymal transition through direct targeting of Ubiquitin Conjugating Enzyme E2 C (UBE2C), thereby controlling cervical cancer cell proliferation through the ubiquitination pathway [59]. Sun et al. described suberoylanilide hydroxamic acid to not only inhibit HDAC1/2/3, thereby promoting mitochondrial-dependent apoptosis but also involve the PTEN-induced putative kinase 1 (PINK1)-Parkin signaling pathway in human cervical cancer cell mitophagy induction in an acetylation-dependent manner [60], whereas Xia et al. found that vorinostat stimulates major histocompatibility class I related chain A (MICA) via the PI3K/Akt pathway and increases cervical cancer cell sensitivity to the natural killer cell-mediated cytolytic reactions [61]. Notably, You et al. first evaluated the effect of suberoyl bishydroxamic acid on the growth of HeLa cells and observed reactive oxygen species (ROS)-independent, glutathione-dependent apoptosis induction [62], while, one year later, the same study group described suberoylanilide hydroxamic acid-induced human cervical cancer cell death to correlate with oxidative stress and levels of thioredoxin 1 [63].

A number of studies spotlight the synergistic effects of suberoylanilide hydroxamic acid with other therapeutic agents, accordingly. Jin et al. applied cisplatin and suberoylanilide hydroxamic acid or sirtinol to HeLa cells, which synergistically activated caspase-3 and induced Bcl-2/x-linked inhibitor of apoptosis protein (XIAP)-mediated apoptosis [64], while Kumar et al. employed disulfide cross-linked biodegradable polymeric nanogels to co-deliver vorinostat and etoposide to HeLa cells and reported synergistic caspase-3/7 activation and enhanced cytotoxicity [65]. Additionally, Jiang et al. examined the effects of the proteasome inhibitor bortezomib and suberoylanilide hydroxamic acid in human cervical cancer cells and noted a significant increase in both caspase-3 stimulation and Bcl-2-associated X (Bax)/Bcl-2 expression, decreased nuclear transportation of nuclear factor-κB (NF-κB), as well as downregulation of Akt expression and phosphorylation [66]. Lange et al. examined the combination of suberoylanilide hydroxamic acid and SBE13 for the treatment of HeLa cervical carcinoma cells and described strongly elevated cell numbers in G2/M and S phase, accompanied by decreasing G0/G1 percentages, polo-like kinase 1 (Plk1) protein reduction, p21 protein induction, as well as caspase-3 and poly (ADP-ribose) polymerase (PARP) activation [67]. Of note, Han et al. combined suberoylanilide hydroxamic acid with the oncolytic adenovirus ZD55-TRAIL and demonstrated synergistic NF-κB/NF-κ light polypeptide gene enhancer in B-cells inhibitor, alpha (IκBα) upregulation, in vitro cell cycle growth arrest and apoptosis induction, as well as in vivo tumor growth reduction [68]. Xing et al. explored the sensitization of vorinostat on chemoradiation for SiHa cells and concluded that suberoylanilide hydroxamic acid in chemotherapy upregulates p21 and Bax, hence promoting apoptosis and cell cycle arrest in G0/G1 phase, while it induces apoptosis and hinders cell repair in radiotherapy via Bax up- and Ku70 downregulation [69].

Taken together, these results indicate that vorinostat may effectively inhibit cervical cancer genesis and progression.

### 2.3. Panobinostat

LBH589 is a potent pandeacetylase inhibitor [70].

In 2016, Wasim et al. investigated the effect of panobinostat on HeLa and SiHa cells and indicated that this HDACI, alone or in combination with topoisomerase inhibitors, causes cell cycle arrest and apoptosis through an increase in the ROS production, the disruption of mitochondrial membrane potential, the reduction of *Bcl-xL* expression, as well as the increase of p21 and caspase-3/7/9 levels [71]. Two years later, the same study group elaborated on the synergistic effects of the combination of panobinostat with topotecan or etoposide and underlined the enhanced apoptosis induction through amplified ROS production and mitochondrial apoptotic pathway induction, along with the downregulation of the PI3K/Akt and NF-κB pro-survival pathways and the activation of the extracellular signal-regulated kinase (ERK) pathway [72].

### 2.4. Abexinostat

Banuelos et al. tested the capacity of the orally bioavailable hydroxamate-based pan-HDACI PCI-24781 to promote radio-sensitization in human cervical carcinoma cells. Exposures longer than 4 h were in general necessary, whereas radiation-resistant S-phase cells required an additional 24-h pretreatment with high abexinostat doses. Notably, cells might block and die in G2/M phases during this period. Even though the abexinostat application compromised DNA repair fidelity, cell cycle redistribution did not constitute a key radio-sensitization mechanism [73].

### 2.5. Thiazole-5-hydroxamic Acid

Anandan et al. connected the thiazole ring with a piperazine spacer capped with a sulfonamide group, thereby designing thiazole-5-hydroxamic acid as a novel HDACI. Importantly, the electron-donating groups on the aryl ring were shown to enhance the potency of these novel HDACIs in a human cervical cancer cell nuclear extract [74].

### 2.6. Amino Benzohydroxamic Acid

Korkmaz et al. generated new small molecular weight amino benzohydroxamic acid derivatives which exhibited potent antiproliferative effects on HeLa cells and suppressed glutathione reductase and thioredoxin reductase (TRXR1) activities [75].

### 2.7. Tubastatin A

Chen et al. treated cervical cancer cell lines with the HDAC6-specific inhibitor tubastatin-A which blocked stromal-interaction molecule 1 (STIM1) trafficking and repressed the activation of STIM1-mediated store-operated Ca^2+^ entry (SOCE) [76].

## 3. Short Chain Fatty Acids

### 3.1. Valproic Acid

Valproic acid is an HDACI approved for the treatment of epilepsy [24,77].

Multiple studies explore the antiproliferative effects of valproic acid as a monotherapy on cervical cancer cell lines. Feng et al. applied valproic acid on three cervical cancer cell lines with distinctive molecular and genetic characteristics and found that this HDACI both stimulated Notch1 cleavage and downregulated E6, thus inducing epithelial-mesenchymal transition. Additional effects included differential proliferation suppression, apoptosis induction, as well as cell cycle arrest [78]. Furthermore, Han et al. evaluated the effects of valproic acid on HeLa cells and indicated that valproic acid may promote apoptosis in a caspase-dependent manner, but independent of changes in ROS and glutathione levels [79]. Rocha et al. proposed methylation status modulation of lysine residues 4, 9, and 27, in histone H3, following treatment with valproic acid in HeLa cells [80], while Zhao et al. investigated the effects of valproic acid on the angiogenesis of cervical cancer and revealed inhibition of PI3K/Akt and ERK1/2 pathways to suppress HIF-1a and VEGF [81]. Sami et al. studied the effects of valproic acid on cervical cancer models, which elevated the expression of p21 and histone H3 acetylation, had a direct anti-angiogenic effect, as well as inhibited tumor growth, and improved survival advantages in vivo [82].

Two different study groups have examined the potency of valproic acid derivatives in cervical cancer cells, as well. Contis-Montes de Oca et al. treated HeLa cells with various concentrations of the valproic acid derivative N-(2′-Hydroxyphenyl)-2-propylpentanamide (OH-VPA) and showed that OH-VPA not only induces high-mobility group box 1 (HMGB1) translocation from the nucleus to the cytoplasm but also boosts ROS levels and displays negative effects on cancer cell viability [83]. Sixto-López et al. synthesized the valproic acid N-(20-hydroxyphenyl)-2-propylpentanamide (HO-AAVPA) derivative which downregulated HDAC1 following nuclear to cytoplasmic HMGB1 translocation and amplified ROS production in human cervical cancer cells [84].

Several studies have been also published on the synergistic effects of valproic acid in combination with other, standard or alternative, therapeutic agents. Li et al. highlighted the enhanced, synergistic, cytotoxic effect of valproic acid and the Aurora kinase inhibitor VE465 on cervical cancer cells [85], whereas Mora-García et al. treated cervical cancer cells with hydralazine/valproate and reported human leukocyte antigen (HLA) class-I expression upregulation, histone H4 hyperacetylation, as well as antigenic immune recognition by cytotoxic T lymphocytes specific to HPV-16/18 E6 and E7-derived epitopes [86]. Moreover, Franko-Tobin et al. suggested that valproic acid suppresses cervical cancer cell growth and promotes the expression of Notch1 and somatostatin receptor subtype 2 (SSTR2), while its combination with the SSTR2-targeting cytotoxic conjugate camptothecin-somatostatin (CPT-SST) correlates with elevated therapeutic potency [87]. Segura-Pacheco et al. supported the addition of valproic acid to adenoviral-mediated cancer gene therapy clinical trials to enhance adenoviral-mediated gene delivery to cervical carcinoma cells via coxsackie adenovirus receptor (CAR) upregulation [88], while Li et al. co-treated human cervical carcinoma cell lines with valproic acid and the oncolytic parvovirus H-1PV, which resulted in the in vitro induction of oxidative stress, DNA damage, and apoptosis, as well as a complete cancer remission in vivo [89]. Mani et al. evaluated the radio-sensitizing effects of hydralazine in combination with valproic acid in SiHa cells and discovered that the triple combination of hydralazine, valproic acid, and cisplatin, correlated with the highest cytotoxic effects in radio-sensitized cervical cancer cells [90]. Zhou et al. developed the new agent Ir-valproic acid which managed to inhibit HDAC6 in HeLa cells and, consecutively, switch the mode of cell death (apoptosis vs. autophagy) [91]. In 2012, Feng et al. combined valproic acid with all-*trans* retinoic acid (ATRA), which led to the hyperacetylation of histone H3. Other effects included the upregulation of *retinoic acid receptor* (*RAR*) *β2*, p21*^CIP1^*, and p53, or the decrease in p-Stat3. The cell cycle was also predominantly arrested at the G1 phase, cancer cell differentiation was promoted, whereas cervical cancer cell growth was halted both in vitro and in vivo [92]. One year later, the same scientific group evaluated the antitumor effects of this combinational treatment in a xenograft model implanted with poorly differentiated human squamous cell carcinoma. Similarly, *RARβ2*, *E-cadherin*, *p21^CIP1^*, and *p53*, were re-expressed, involucrin and loricrin experienced upregulation, whereas p-Stat3 was reduced [93].

Most significantly, certain studies describe valproic acid’s efficacy in cervical cancer tumor samples or even clinical trials. De la Cruz-Hernández et al. concluded that valproate, either alone or in combination with hydralazine, exerts a growth inhibitory effect on cervical cancer cell lines partially due to the valproate-induced hyperacetylation of p53 protein (which is otherwise degraded by E6) [94]. In 2011, the same research group analyzed 10 pairs of pre- and posttreatment cervical tumor samples by microarray analysis and stated that combinational hydralazine/valproate application in patients upregulated 964 genes associated with energy production and, by extension, tumor suppression, as well as increased acetylated p53 [95]. After a baseline tumor biopsy and blood sampling, Chavez-Blanco et al. treated 12 women diagnosed with cervical carcinoma with magnesium valproate and evaluated tumor acetylation of H3 and H4 histones along with HDAC activity by Western blot and colorimetric HDAC assay, respectively. Interestingly, magnesium valproate at doses between 20–40 mg/kg was found to inhibit HDAC activity and to hyperacetylate histones in cervical cancer tissues [96]. A year later, the same study group treated HeLa cells with a DNA methylation inhibitor and the HDACI plus chemotherapeutic agents and noted statistically significant higher cytotoxicity after combining hydralazine and valproic acid with cisplatin, adriamycin, or gemcitabine [97]. In a double-blind, placebo-controlled, randomized phase III trial of hydralazine/valproate plus cisplatin/topotecan, Coronel et al. outlined the superiority of the epigenetic therapy over standard combination chemotherapy in terms of progression-free survival (PFS) in advanced cervical cancer, at expenses of slightly elevated but still acceptable toxicity [98].

Altogether, valproic acid, alone or in combination with other drugs, represents a potent HDACI in the context of cervical cancer therapy.

### 3.2. Sodium Butyrate

Sodium butyrate is a part of the metabolic fatty acid fuel cycle that also acts as an HDACI [99].

Darvas et al. treated HPV18-positive human cervical cancer cells with sodium butyrate, which suppressed the transcription of cellular FADD-like IL-1β-converting enzyme (FLICE)-inhibitory protein (c-FLIP), hindered caspase-8 recruitment and, consequently, augmented HeLa cell sensitivity to tumor necrosis factor (TNF) α and TNF-related apoptosis-inducing ligand (TRAIL)-induced apoptosis. Of note, solely continued viral oncogene expression favored sensitization, given that short-interfering RNA (siRNA)-mediated HPV18 E6/E7 transcription knockdown prevented HDACI/death-ligand mediated apoptosis [100]. Finzer et al. cultured cervical cancer cells and stated that sodium butyrate not only induced cell cycle arrest in HPV-18 positive cervical carcinoma cells, but also modulated cyclins and induced cyclin-dependent kinase (CDK) inhibitor p21*^CIP1^* on the transcriptional level. CDK2 activity was also found to be suppressed, while pRb was degraded in an E7-dependent manner [101]. Two years later, the authors published a second paper on the derivative phenylbutyrate. More accurately, phenylbutyrate application induced the CDK inhibitors p21*^CIP1^* and p27*^KIP1^* and caused a dose-dependent cervical cancer cell growth arrest by augmenting the G1 fraction [102].

Some studies also focus on the synergistic effects of sodium butyrate with diverse therapeutic agents. Bachmann et al. pretreated HeLa cells with sodium butyrate prior to TNF-a addition and, surprisingly, reported a lack of *interferon* (*IFN*)- *β* gene inducibility [103]. Additionally, Decrion-Barthod et al. combined sodium butyrate with 7-hydroxystaurosporine (UCN-01), which led to increased caspase-3 and poly (ADP-ribose) polymerase cleavage in vitro after p21, Bcl-2, Bcl-2-associated X protein, and B-cell lymphoma-extra large (Bcl-xL) modulation. In vivo, the aforementioned agent combination substantially suppressed cervical cancer growth, accordingly [104]. Park et al. treated HeLa cells with sodium butyrate in combination with wortmannin or LY294002 and pointed out enhanced caspase-3/9 and PARP activation, cell cycle arrest, upregulation of p21^Cip1/Waf1^ and p27^Kip1^, as well as a reduction in the expression levels of cyclins A, B1 and D1 [105].

Taken together, these results imply that sodium butyrate might successfully hinder cervical cancer genesis and progression.

### 3.3. Phenylbutyrate

Almotaity et al. incorporated in axial position the HDACI 4-phenylbutyrate in platinum (IV) derivatives of carboplatin and found compound *cis-trans*-[Pt(CBDCA)(NH3)2(PBA)(bz)] to display higher efficacy than the standard chemotherapeutic agent and to have a negative effect on cellular basal HDAC activity in A431 human cervical cancer cells [106].

## 4. Benzamides

### Domatinostat

Zhang et al. investigated the role of the selective class I HDACI 4SC-202 in cervical cancer cells and reported cell cycle arrest in the G2/M phase arrest, apoptosis induction, inhibition of the prolactin receptor-related pathways, as well as tumor growth restriction in vivo [107].

## 5. Cyclic Tetrapeptides

### 5.1. Romidepsin

Song et al. established the immunoresistant CaSki P3 cervical tumor cell line and suggested that romidepsin may exhibit more potent antitumor effects upon HDAC activation via immune selection with antigen-specific T cells [108].

### 5.2. Apicidin

Apicidin is a fungal metabolite exerting antiparasitic activity by the inhibition of HDAC [109].

Łuczak et al. performed quantitative real-time polymerase chain reaction (qRT-PCR) and Western blot analysis and observed that both HPV16-E6 and -E7 transcripts and protein levels experience significant downregulation after apicidin application to SiHa cells [110]. Furthermore, You et al. reported that apicidin treatment diminishes DNA methyltransferase 1 expression and incites repressive histone modifications on the promoter sites of HeLa cells [111]. Inspired by the observation that gelsolin may generally exert anticancer effects, Eun et al. showed that apicidin mediates gelsolin induction in HeLa cells via the protein kinase C (PKC) signaling pathway, and especially the PKCε isoform [112].

## 6. Sirtuin Inhibitors

Sirtuins are classified as class III HDACs [113].

Kuhlmann et al. demonstrated that treatment with a Rho guanine nucleotide dissociation inhibitor α (RhoGDIα)-derived K52-trifluoroacetylated, substrate-derived peptidic sirtuin (SIRT) inhibitor harshly hinders cervical cancer cell proliferation [114]. Furthermore, Singh et al. noted that the SIRT1 inhibitor Ex527 and the SIRT2 inhibitor AGK2 impair the growth of SiHa cells, with SIRT1 inhibition causing cell death and SIRT2 inhibition resulting in cell cycle arrest [115]. Wössner et al. revealed thiocyanates as novel selective SIRT1 inhibitors which provoked hyperacetylation of p53 and H3 and inhibited tumor cell proliferation, migration, or colony formation, in HeLa cells [116].

## 7. Novel Synthetic HDACIs

Borutinskaite et al. examined the antiproliferative effect of retinoic acid and HDACI BML-210 on HeLa cells and highlighted that the aforementioned combination caused a marked increase in p21, Bcl-2, and phosphorylated p38 mitogen-activated protein (MAP) kinase levels, thereby regulating cell cycle arrest and inducing apoptosis of HeLa cells [117]. In 2012, the same study group published their second research article on this topic and indicated that HeLa cell treatment with BML-210 and retinoic acid downregulated HDAC classes I and II and determined changes in protein expression levels of dystrophin, neuronal nitric oxide synthase (NOS1), as well as different isoforms of dystrobrevin, hence inhibiting cell growth and inducing apoptosis in cervical carcinoma cells [118].

In 2014, Ravichandiran et al. synthesized a new series of 2-(4-aminobenzosulfonyl)-5H-benzo[b]carbazole-6,11-dione derivatives which exhibited good cytotoxic activity as HDAC8 inhibitors in SiHa cells [119].

## 8. Statins

Lin et al. examined the effect of the combination of statins with various HDACIs and demonstrated that fluvastatin and lovastatin induce p21 expression in a p53-independent manner, as well as selectively cooperate with numerous HDACIs in HeLa cells through pathways related to apoptosis, autophagy, cell cycle progression or DNA damage [120].

## 9. Pyruvate/Lactate

With a view to exploring the role of pyruvate as HDACI in cervical cancer, Ma et al. used tumor samples and noted that intracellular pyruvate concentrations are inversely related to histone protein levels [121]. Furthermore, Wagner et al. examined the role of lactate in the DNA damage response of HeLa cells and reported HDAC inhibition, alongside histone H3/4 hyperacetylation, which resulted in a decrease in chromatin compactness [122].

## 10. Phytopharmaceuticals

A large number of studies have explored the role of diverse phytopharmaceuticals as efficient HDACIs in cervical cancer.

Anantharaju et al. screened for HDAC inhibitory effect using in silico docking methods and identified cinnamic acids as potential naturally occurring HDACIs. More precisely, caffeic acid better interacted with HDAC2 in silico and decreased its activity both ex vivo and in vitro, hence inducing cancer cell death by ROS generation, cell cycle arrest in S and G2/M phases, as well as caspase-3 mediated apoptosis induction [123].

An Indian study group checked whether the ethanolic extract of Gonolobus condurango Condurango 30C was capable of arresting the cell cycles in HeLa cervical cancer cells and underlined that this homeopathically-diluted anti-cancer remedy induced cytotoxicity in vitro, reduced DNA synthesis, promoted G1-phase cell-cycle arrest, as well as downregulated HDAC2 activity [124].

Saenglee et al. found peanut phenolics to possess HDAC inhibitory capacities and to induce apoptosis and cell cycle arrest in HeLa cells [125].

Moreover, Senawong et al. treated HeLa cells with Hydnophytum formicarium Jack. rhizome ethanolic and phenolic-rich extracts and observed that these extracts, together with sinapinic acid, induced cervical cancer cell apoptosis [126].

Roy et al. explored the effect of curcumin on SiHa cells and reported that both HDAC1/2 and HPV inhibition resulted in cell cycle arrest at the G1/S phase by alteration of cell cycle regulatory proteins [127], while Ramnath et al. revealed that the terpenoid abietic acid was docked with HDAC3 receptors in HeLa cells, hence inducing apoptosis in a concentration-dependent manner [128].

Mustafa et al. identified the sulfated polysaccharide fucoidan as a novel HDACI that induces ROS-dependent epigenetic modulation in HeLa cells [129].

Interestingly, Phaosiri et al. tested 20 newly synthesized derivatives of [6]-shogaol (4) for inhibitory activity against HDACs in HeLa cells and identified compound 5j as the most efficient agent in terms of antiproliferative activity [130].

Raina et al. applied chrysin to HeLa cells and reported dose-dependent epigenetic alterations in terms of HDAC1/2/3/4/11 expression [131], whereas Mazzio et al. conducted high-throughput screening of non-fermented nutraceuticals and food-based polyphenolics and showed that, in HeLa cells, 13 plant-based HDACIs resemble trichostatin A by promoting the expression of microRNAs responsible for tumor suppression [132].

In 2015, Khan et al. investigated the role of both the green tea catechin (-)-epigallocatechin-3-gallate (EGCG) and sulforaphane as epigenetic modifiers in HeLa cells, which were found to decrease the expression and inhibit the activity of HDAC1 [133].

Sundaram et al. examined genistein’s role as a natural HDACI in HeLa cells and noted a significant reduction of the expression and enzymatic activity of HDACs in a time-dependent manner [134]. In 2019, they also announced that the phytochemical quercetin decreases the activity of HDACs in a dose-dependent manner via interaction with residues in their catalytic cavities [135]. One year later, the same study group examined the effect of EGCG on HeLa cells and concluded that it may competitively inhibit HDAC2, HDAC3, HDAC4, and HDAC7, thereby interfering with promoter hypermethylation and transcriptional expression of tumor suppressor genes [136].

Altogether, phytopharmaceuticals seem to represent novel natural HDACIs with great potency in cervical cancer.

Table 2 and Table 3 summarize the major effects of the most studied ‘conventional’ HDACIs and phytopharmaceuticals as monotherapies on cervical cancer, respectively.

Figure 3 and Figure 4 schematically explain the mechanism of action of different HDACIs on cervical cancer.

Table 4 summarizes the possible combinations of HDACIs with other, standard or alternative, treatment agents in the context of cervical cancer therapy.

## 11. Discussion

Cervical cancer represents the fourth most common cancer entity among women worldwide [137]. Localized cervical carcinomas correlate with a 5-year relative survival rate of 92%. Nonetheless, the five-year survival rates for advanced cervical cancer patients are discouraging and the percentage of patients diagnosed with invasive cervical cancer in a distant Surveillance, Epidemiology, and End Results (SEER) stage only reaches 17% [138]. Consequently, the much-needed amelioration of patient overall survival (OS), disease-free survival (DFS), and PFS, undoubtedly necessitates the timely generation of novel efficient anticancer drugs.

The present review of the literature comprehensively summarizes the results of all currently available original research articles on the role of HDACIs in terms of cervical cancer therapy. Both well-established and novel HDACIs seem to represent innovative, potent anticancer therapeutic agents, which, either alone or in combination with other treatments, may inhibit cervical cancer cell growth, provoke cell cycle arrest, and induce apoptosis. In this context, the most studied HDACIs seem to be trichostatin A, vorinostat, valproic acid, as well as sodium butyrate. All four HDACIs were found to effectively inhibit cervical cancer genesis and progression, with a significantly multiplied efficacy once combined with standard chemotherapeutics, other alternative treatment agents, or irradiation. This observation is of utmost importance, given that HDACIs may even modernize and improve already approved treatment modalities with sufficiently documented effectiveness and identified side effects. Except for the ‘traditional’ HDACIs, newly designed drugs also seem to exert respective antitumor effects on cervical cancer cells, whereas the identification of agents such as statins as HDACIs paves the way for exploring and hopefully discovering the hidden additional features of meanwhile ‘tedious’ drug substances. Interestingly enough, phytopharmaceuticals have been also reported to act as potent HDACIs in cervical cancer, thereby highlighting the usefulness of natural products and undoubtedly reviving the hope for several cervical cancer patients who do not wish to receive standard chemo- and/or radiotherapy.

The majority of the presented works employed the HeLa or SiHa cervical cancer cell lines to test the therapeutic properties of HDACIs in cervical cancer cells, while only a few study groups investigated the effectiveness of HDACIs in cervical cancer xenograft models. Only two research groups conducted clinical trials and published the most promising results concerning the application of valproic acid (in combination with other treatment agents) in cervical carcinoma patients. Consequently, future studies could incorporate further cervical cancer cell lines but should also take into consideration the advantages of the employment of more reliable, three-dimensional testing models including organoids, which have so far been described as the most useful drug sensitivity screening platforms in different types of (gynecological) cancer [139,140,141]. Preferably, the conduction of clinical trials with large patient collectives could help verify the clinical utility and safety of diverse HDACIs in cervical cancer therapy, as well as inspect eventual adverse side effects resulting from their application to women.

## 12. Conclusions

In summary, HDACIs may pave the way for the establishment of novel, effectual therapeutic agents, which may re-determine, or even replace, standard radio chemotherapeutic regimens, and offer better treatment alternatives for women suffering from cervical cancer. The mechanism of action is, nevertheless, complex and, therefore, difficult to implement in clinical practice. Moreover, the potential synergistic or replacing role of HDACIs definitely requires further discussion, especially in a time of evolving immunotherapy in cervical cancer.

## Figures and Tables

**Figure 1 cancers-15-02222-f001:**
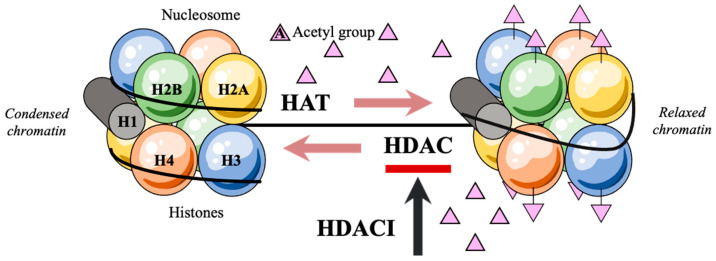
Post-translational acetylation decreases the affinity of histones for the negatively charged DNA, hence allowing DNA strands to uncoil and transcription to occur.

**Figure 2 cancers-15-02222-f002:**
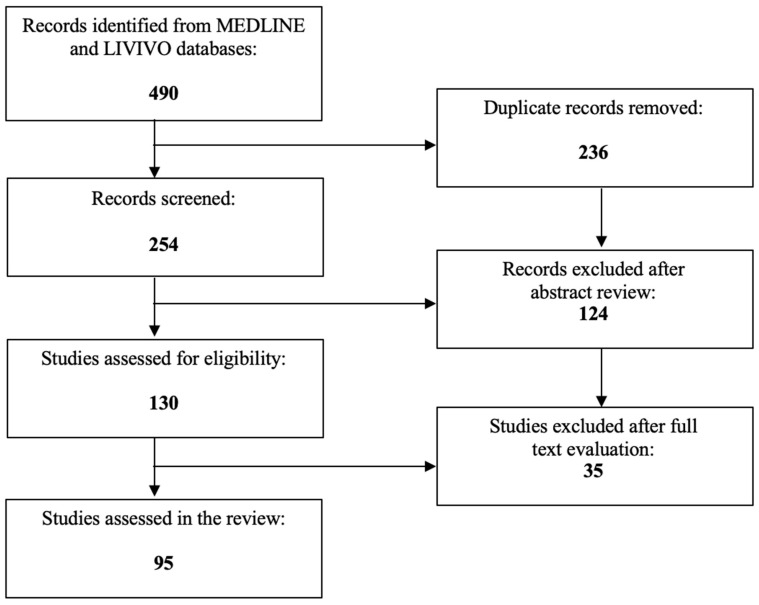
PRISMA flow diagram visually summarizing the screening process.

**Figure 3 cancers-15-02222-f003:**
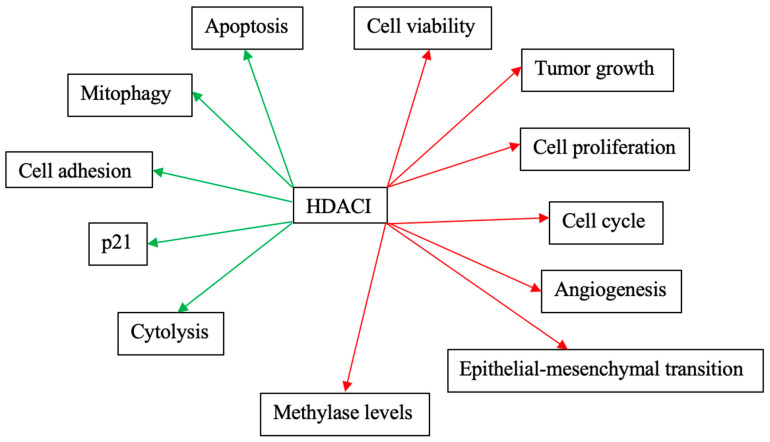
Mechanisms of action of the most studied HDACIs as monotherapies on cervical cancer. Red arrows: Inhibitory effects. Green Arrows: Activating effects.

**Figure 4 cancers-15-02222-f004:**
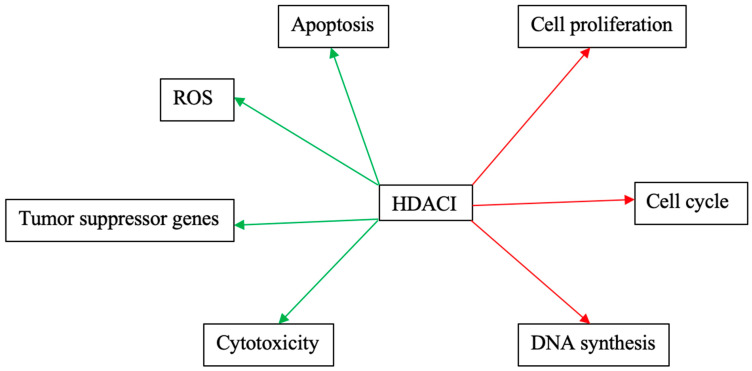
Mechanisms of action of phytopharmaceuticals as monotherapies on cervical cancer. Red arrows: Inhibitory effects. Green Arrows: Activating effects.

**Table 1 cancers-15-02222-t001:** Overview of selected HDACIs.

HDACI	Class	Target HDAC Class
Hydroxamic acids	Trichostatin A	Pan
	Suberoylanilide hydroxamic acid	Pan
	Panobinostat	Pan
	Abexinostat	Pan
Short-chain fatty acids	Valproic Acid	I, IIa
	Sodium butyrate	I, II
	Phenylbutyrate	I, II
Benzamides	Domatinostat	I
Cyclic peptides	Romidepsin	I
	Apicidin	I

**Table 2 cancers-15-02222-t002:** Major effects of the most studied HDACIs as monotherapies on cervical cancer.

HDACI	Major Effects on Cervical Cancer	References
Trichostatin A	Apoptosis induction Decreased methyltransferase levels Cell viability impairment p21 activation Improved cell adhesion In vivo tumor growth inhibition	[34,35,36,37,38,39,40,41,42,43,44]
Vorinostat	Reversed epithelial-mesenchymal transition Apoptosis and mitophagy induction Sensitization to natural killer cell-mediated cytolytic reactions	[57,58,59,60,61,62,63]
Valproic acid	Cell proliferation suppression Apoptosis induction Cell cycle arrest Epithelial-mesenchymal transition induction Angiogenesis suppression In vivo tumor growth inhibition	[78,79,80,81,82]
Sodium butyrate	Apoptosis induction Cell cycle arrest	[100,101,102]

**Table 3 cancers-15-02222-t003:** Major effects of phytopharmaceuticals as monotherapies on cervical cancer.

Phytopharmaceutical	Major Effects on Cervical Cancer	References
Caffeic acid	Apoptosis induction Cancer cell death by ROS generation Cell cycle arrest	[123]
Condurango 30C	Cell cycle arrest Cytotoxicity induction DNA synthesis reduction	[124]
Peanut phenolic	Apoptosis induction Cell cycle arrest	[125]
Hydnophytum formicarium Jack. rhizome extract	Apoptosis induction	[126]
Curcumin	Cell cycle arrest	[127]
Terpenoid abietic acid	Apoptosis induction	[128]
[6]-shogaol (4)	Antiproliferative activity	[130]
EGCG	Interaction with promoter hypermethylation and transcriptional expression of tumor suppressor genes	[136]

**Table 4 cancers-15-02222-t004:** Possible combinations of HDACIs with other treatment agents for synergistic cervical cancer therapy.

HDACI	Synergistic Agent	References
Trichostatin A	5-Aza-dC Alkylating agents Demethylation inducers Pyruvate Vorinostat Bortezomib Curcumin Silver nanoparticles Palladium nanoparticles Radiation	[45,46,47,48,49,50,51,52,53,54]
Vorinostat	Cisplatin Etoposide Bortezomib SBE13 ZD55-TRAIL Radiation	[64,65,66,67,68,69]
Panobinostat	Topoisomerase inhibitors Topotecan Etoposide	[71,72]
Abexinostat	Radiation	[73]
Valproic acid	VE465 Hydralazine CPT-SST Adenovirus H-1PV Cisplatin Adriamycin Gemcitabine Topotecan Retinoic acid Radiation	[85,86,87,88,89,90,91,92,93,94,95,97,98]
Sodium butyrate	UCN-01 Wortmannin LY294002	[103,104,105]
BML-210	Retinoic acid	[117,118]

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
