# Peer review of "The Emerging Role of Histone Deacetylase Inhibitors in Cervical Cancer Therapy"

_cancers, 2023, doi:10.3390/cancers15082222_

Round 1
Reviewer 1 Report
Review Cancers-2324025: The Emerging role of Histone Deacetylase Inhibitors in Cervical Cancer Therapy.
Dear Editor, thank you for the opportunity to read above mentioned manuscript in advance. Authors provide a comprehensive review about current status of research and possible clinical indications of histone deacetylase inhibitors (HDACIs) in cervical cancer. My comments are the followings:
1. This is a meticulous list of available publications in this oncologic field and a summary of condensed summaries without any critical judging. Indeed this I my major point of criticism.
2. Page 2 line 69: I miss an adenocarcinoma in situ as precursor lesion of adenocarcinoma.
3. Page 2 line 76-80: Surgical treatment reaches from conisation via radical trachelectomy/hysterectomy up to exenteration. Standard treatment for locally advanced cervical cancer is platinum-based chemoradiation and for primary metastatic disease primary chemotherapy (+/- Immuntherapy).
4. In general I miss a statement that mechanism of action is complex and therefore difficult to implement in clinical practice. Moreover, the potential synergistic or replacing role of HDACIs definitely needs more discussion. In a time of evolving immunotherapy in cervical cancer a longer hint to this group of drugs is mandatory.
Author Response
Review Cancers-2324025: The Emerging role of Histone Deacetylase Inhibitors in Cervical Cancer Therapy.
Dear Editor, thank you for the opportunity to read above mentioned manuscript in advance. Authors provide a comprehensive review about current status of research and possible clinical indications of histone deacetylase inhibitors (HDACIs) in cervical cancer. My comments are the followings:
- This is a meticulous list of available publications in this oncologic field and a summary of condensed summaries without any critical judging. Indeed, this I my major point of criticism.
In the Discussion and Conclusions part of the review, we tried to comprehensively summarize and critically judge the results of the different publications.
- Page 2 line 69: I miss an adenocarcinoma in situ as precursor lesion of adenocarcinoma.
We have now added adenocarcinoma in situ as precursor lesion of adenocarcinoma.
- Page 2 line 76-80: Surgical treatment reaches from conisation via radical trachelectomy/hysterectomy up to exenteration. Standard treatment for locally advanced cervical cancer is platinum-based chemoradiation and for primary metastatic disease primary chemotherapy (+/- Immuntherapy).
We have now added these important points in the Introduction part of the review.
- In general I miss a statement that mechanism of action is complex and therefore difficult to implement in clinical practice. Moreover, the potential synergistic or replacing role of HDACIs definitely needs more discussion. In a time of evolving immunotherapy in cervical cancer a longer hint to this group of drugs is mandatory.
Thank you for this useful comment. We have now made the suggested statement in the Discussion and Conclusions part of the review.
Reviewer 2 Report
This is a well-written manuscript titled “The Emerging Role of Histone Deacetylase Inhibitors in Cervical Cancer Therapy” that discusses all studies with HDAC inhibitors as treatment agents for cervical cancer. The figures/ tables/ schemes are appropriate and align with the text and in the present review, the author has comprehensively summarized the results of all 95 studies published between 2001 and 2023 on the role of HDACIs for cervical cancer therapy. This manuscript also identified a gap in knowledge and will be of good interest to the scientific community. The manuscript can be accepted in its present form with minor revisions.
1. The figures/ tables/schemes are self-explanatory and nicely illustrated in the present review. It will be helpful for the reader if the author adds some figures explaining the mechanism of action of different HDAC inhibitors on cervical cancer.
2. In this review the majority of the studies employed the HeLa or SiHa cervical cancer cell lines to test the therapeutic efficacy of HDACIs in cervical cancer cells, but few studies on the effectiveness of HDACIs in cervical cancer xenograft models. If any studies available on organoid Cervical cancer and patient-derived cell lines models for therapeutic efficacy of HDAC inhibitors should be of great interest to the author.
Author Response
This is a well-written manuscript titled “The Emerging Role of Histone Deacetylase Inhibitors in Cervical Cancer Therapy” that discusses all studies with HDAC inhibitors as treatment agents for cervical cancer. The figures/ tables/ schemes are appropriate and align with the text and in the present review, the author has comprehensively summarized the results of all 95 studies published between 2001 and 2023 on the role of HDACIs for cervical cancer therapy. This manuscript also identified a gap in knowledge and will be of good interest to the scientific community. The manuscript can be accepted in its present form with minor revisions.
- The figures/ tables/schemes are self-explanatory and nicely illustrated in the present review. It will be helpful for the reader if the author adds some figures explaining the mechanism of action of different HDAC inhibitors on cervical cancer.
Thank you for this useful suggestion. We have now added Figures 3 and 4 explaining the mechanism of action of different HDAC inhibitors on cervical cancer.
- In this review the majority of the studies employed the HeLa or SiHa cervical cancer cell lines to test the therapeutic efficacy of HDACIs in cervical cancer cells, but few studies on the effectiveness of HDACIs in cervical cancer xenograft models. If any studies available on organoid Cervical cancer and patient-derived cell lines models for therapeutic efficacy of HDAC inhibitors should be of great interest to the author.
Thank you for this useful comment. The present review summarizes and discusses all studies with HDAC inhibitors as treatment agents for cervical cancer. Unfortunately, we were not able to find any relevant studies employing cervical cancer organoids or patient-derived cell lines models, which would have, indeed, been of great interest to the readers.